# Spontaneous Fusion with Transformed Mesenchymal Stromal Cells Results in Complete Heterogeneity in Prostate Cancer Cells

**DOI:** 10.3390/cancers16050951

**Published:** 2024-02-27

**Authors:** Ruoxiang Wang, Peizhen Hu, Fubo Wang, Ji Lyu, Yan Ou, Mouad Edderkaoui, Yi Zhang, Michael S. Lewis, Stephen J. Pandol, Haiyen E. Zhau, Leland W. K. Chung

**Affiliations:** 1Department of Medicine, Cedars-Sinai Medical Center, Los Angeles, CA 90048, USA; peizhen.hu@cshs.org (P.H.); fubo.wang@cshs.org (F.W.); ji.lyu@cshs.org (J.L.); mouad.edderkaoui@cshs.org (M.E.); stephen.pandol@cshs.org (S.J.P.); haiyen.zhau@cshs.org (H.E.Z.); leland.chung@cshs.org (L.W.K.C.); 2Department of Biomedical Sciences, Cedars-Sinai Medical Center, Los Angeles, CA 90048, USA; yan.ou@cshs.org (Y.O.); yi.zhang@cshs.org (Y.Z.); 3Department of Medicine and Pathology, Cedars-Sinai Medical Center, Los Angeles, CA 90048, USA; michael.lewis3@cshs.org

**Keywords:** tumor cells, bystander cells, stromal cell transformation, cell fusion, cancer–stromal hybridization, heterogeneity progression

## Abstract

**Simple Summary:**

We have previously demonstrated that the interaction between cancer cells and bystander cells in the tumor microenvironment can promote the development of tumor cell heterogeneity. This heterogeneity is the basis for cancer progression, therapeutic resistance, and distant metastasis. In this study, we co-cultured prostate cancer cells with stromal cells to study the process of cancer–bystander cell interaction. Compared to primary stromal cells, spontaneously transformed stromal cells had significantly enhanced activity in inducing heterogeneity in cancer cells. By tracking the individual cancer–stromal interactions, we found that the transformed stromal sublines had a high tendency to fuse with cancer cells, forming cancer–stromal fusion hybrids that had the ability to proliferate. Hybridization between individual cancer cells and stromal cells resulted in genotypic and phenotypic diversification. This study indicates that the pathophysiologic status of the bystander cell compartment is crucial to the progression of tumor cell heterogeneity.

**Abstract:**

Tumor cells gain advantages in growth and survival by acquiring genotypic and phenotypic heterogeneity. Interactions with bystander cells in the tumor microenvironment contribute to the progression of heterogeneity. We have shown that fusion between tumor and bystander cells is one form of interaction, and that tumor–bystander cell fusion has contrasting effects. By trapping fusion hybrids in the heterokaryon or synkaryon state, tumor–bystander cell fusion prevents the progression of heterogeneity. However, if trapping fails, fusion hybrids will resume replication to form derivative clones with diverse genomic makeups and behavioral phenotypes. To determine the characteristics of bystander cells that influence the fate of fusion hybrids, we co-cultured prostate mesenchymal stromal cell lines and their spontaneously transformed sublines with LNCaP as well as HPE-15 prostate cancer cells. Subclones derived from cancer–stromal fusion hybrids were examined for genotypic and phenotypic diversifications. Both stromal cell lines were capable of fusing with cancer cells, but only fusion hybrids with the transformed stromal subline generated large numbers of derivative subclones. Each subclone had distinct cell morphologies and growth behaviors and was detected with complete genomic hybridization. The health conditions of the bystander cell compartment play a crucial role in the progression of tumor cell heterogeneity.

## 1. Introduction

Tumor cell heterogeneity is believed to play a significant role in the progression, metastasis, recurrence, and resistance to anti-tumor therapies [1,2,3]. It is thought that tumors develop from normal cells, which are made up of homogeneous subpopulations with specific functions and limited lifespans [4]. As the tumor progresses, the cells strive to acquire genotypic and phenotypic heterogeneities. Once the heterogeneity reaches a certain degree, the tumor becomes untreatable because it can now produce a wide range of progenies, some of which will always have advantages in surviving therapeutic insults. Elucidating the driving mechanism for heterogeneity progression may help us understand the causes of tumor progression and metastasis.

How do tumor cells become heterogeneous, specifically prostate tumor cells, which are somatic cells with an endodermal epithelium origin? How do they acquire phenotypic traits typically seen in lineages of mesoderm or the neural crest of the ectoderm? Two theories attempt to address the progression of heterogeneity, but neither fully explains the histopathologic changes observed in clinical tumors.

According to the mutator phenotype theory [5,6], tumor cells diversify from each other through successive mutations. However, it is now known that different cells in a tumor have highly heterogeneous mutations with no consensus or cumulative pattern [7,8]. On the other hand, the cancer stem cell theory [9] suggests that heterogeneity arises from the differentiation of pluripotent cancer stem cells upon epigenetic regulation. This theory accounts for the self-renewal perpetuation and lineage plasticity [10] of cancer cells. However, differentiation may lead to pleomorphism but not the complex aneuploidy and genomic heterogeneity seen in tumors [11,12]. The acquisition and progression of tumor cell heterogeneity are complex topics, and the underlying mechanisms are yet to be fully understood.

Alternatively, tumor cell heterogeneity can be induced by interacting with bystander cells, which are non-malignant cells in the tumor microenvironment. These bystander cells include mesenchymal stem/stromal cells, immune cells, endothelial cells, and neural cells, all of which cohabitate with invading tumor cells [13]. The human prostate cancer LNCaP cells, which are androgen-dependent and non-tumorigenic, can become heterogeneous when co-inoculated [14] or co-cultured [15] with non-tumorigenic bone marrow-derived mesenchymal stem/stromal cells. This interaction results in the formation of LNCaP-lineage subclones with distinct genomic changes, androgen-independence, and acquired tumorigenicity. Concurrently, the interaction can convert adjacent bystander cells to malignant cells. Although isogeneity with cancer cells limits identification in clinical tumor specimens, the transformation of bystander cells in the tumor microenvironment can be readily detected in xenograft tumors, as stromal [16,17] or endothelial [18] cells of a mouse host being recruited and reprogrammed into malignant cells.

In our study, we conducted co-cultures and found that LNCaP cells were fusogenic, capable of spontaneously fusing with bystander cell types such as bone marrow- or prostate-derived mesenchymal stem/stromal cells [19] or neural stem cells [20]. Most of these fusion hybrids died during the heterokaryon or synkaryon stage, but a few survived to proliferate into subclones with genotypic and phenotypic diversification. Cancer–bystander cell fusion, therefore, has the potential to introduce heterogeneity into the tumor cell population. Since cancer and bystander cells are known to co-evolve [21], the spontaneity of fusogenicity signifies the reality of an incessant and dynamic heterogeneity progression, which is perhaps the most concerning outcome of cancer cell interaction with the tumor microenvironment.

Through previous studies, we have realized that bystander cells in fusion may have opposite functions. They can either phase out cancer cells by trapping them in the heterokaryon or synkaryon stage or promote tumor cell heterogeneity if the hybrids re-enter the cell cycle. Cancer–bystander cell fusion, therefore, plays intermediating roles in cancer progression and metastasis.

In this current study, we conducted additional co-cultures to investigate how bystander cells could promote the progression of tumor cell heterogeneity. We used the non-tumorigenic LNCaP [19,20] and HPE-15 [22] human prostate cancer cell lines as study subjects and focused on tracking the heterogeneity progression of epithelial cell-like and non-tumorigenic HPE-15 prostate tumor cells through co-culture with prostate mesenchymal stromal cells. Compared to the normal parental control, spontaneously transformed stromal cells showed a significantly enhanced tropism toward HPE-15 heterogeneity progression. The pathophysiological status of bystander cells, therefore, is a determining factor in cancer progression and metastasis.

## 2. Materials and Methods

### 2.1. Cells and Cell Culture

We have previously reported the origin of LNCaP cells [19] and the tagging of these cells with an AsRed2 red fluorescence protein (RFP) in co-culture with bystander cells [19,20]. In this study, we used LNCaP^RFP^, a clone of the tagged LNCaP cells. We also reported the isolation and characterization of the HPE-15 prostate cancer cell line, along with the RFP-tagging and cloning by limiting dilution [22]. For this study, we used HPE-15^RFP^, a clone of the tagged cells between passages 62 and 65 [22].

For the bystander cells, we used six pair-matched human prostate mesenchymal stromal cell lines (HPS-10, HPS-11, HPS-12, HPS-13, HPS-14, and HPS-15). These cell lines were isolated and established from the normal and tumor zones of surgical tumor resections of three prostate cancer patients [23]. We used stromal cells between passages 31 and 35, as cells at higher passages would lose proliferative activity [23]. Additionally, we used PrSC, a normal human prostate stromal cell line from Lonza Bioscience (Walkersville, MD, USA) between passages 31 and 35.

All cells were cultured in T-medium (Formula LS0020056DJ, Life Technologies, Carlsbad, CA, USA) supplemented with 10% fetal bovine serum (FBS, Atlanta Biologicals, Flowery Branch, GA, USA), penicillin (100 units/mL), and streptomycin (100 μg/mL) in a humidified incubator at 37 °C in atmospheric air supplemented with 5% CO_2_.

In later studies, to track bystander cells in co-culture, the stromal cells were tagged with green fluorescence protein (GFP) by infection with TurboGFP lentiviral particles (SHC003V, Sigma-Aldrich, St. Louis, MO, USA) as we previously reported [19]. Clones with GFP expression were isolated by antibiotic selection and limiting dilution. For each stromal cell line, we used a representative clone in this study.

### 2.2. Cell Proliferation Assay

Cells in a single-cell suspension were counted with a TC20 automatic counter (Bio-Rad, Hercules, CA, USA). Equal numbers of cells were cultured in triplicate on 24-well plates (CytoOne, USA Scientific, Ocala, FL, USA) for 7 days. Cell proliferation was assessed every 2 days by crystal violet staining, as we described previously [23].

### 2.3. Co-Culture

We have previously reported the co-culture protocol [19]. In this study, prostate stromal cells were plated onto 6-well plates (CytoOne) and allowed to grow until they formed a stromal monolayer, reaching approximately 80% confluence. Then, about 2.5 × 10^4^ HPE-15^RFP^ cells in a single-cell suspension were overlaid onto the monolayer, resulting in a co-culture of equal numbers of cancer and stromal cells in 4 mL of medium per well. The co-culture was maintained for 8 weeks with weekly medium changes.

### 2.4. Cloning and Continuous Passaging

Mixed cells from a 28-day co-culture were subjected to low density replating (1:300). Specifically, 20% of the cells from a well on a 6-well plate were diluted into four 15-cm culture dishes. After 2 weeks of culture, the colonies that were well-spaced were picked with cloning discs (Bel-Art, Wayne, NJ, USA). These colonies were then grown in 6-well plates for continuous culture with a replating ratio of 1:5.

### 2.5. Genotyping Analysis

We have previously described our protocol for short tandem repeat (STR) genotyping [19]. In brief, after 60 passages of continuous culture, 1 × 10^6^ cells from each individual clone were submitted for genotyping with the commercial Human STR Profiling Cell Authentication Service (ATCC, Manassas, VA, USA). The uniqueness of the clones was confirmed by comparing their STR profiles with the ATCC database. All human cell lines were authenticated by STR profiling within the past three years.

### 2.6. Assessment of Tumorigenic Potential

The protocol for xenograft tumor formation has previously been reported [20,22]. In brief, local tumor formation was assessed by inoculating 2 × 10^6^ cells/site subcutaneously on both flanks of 6-week-old male NCr^nu/nu^ mice (National Cancer Institute, Frederick, MD, USA; *n* = 5). Tumor formation was measured with a caliper biweekly. Tumor volumes were calculated with the formula a^2^ × b × 0.5236 where a was the smallest diameter and b was the opposing diameter. The humane endpoint was reached when the tumor volume reached 1.5 cm^3^, or hemorrhagic tumor ulceration occurred.

### 2.7. Fluorescence Microscopy

The protocol for fluorescence microscopic imaging has previously been reported [19]. For comparison purposes, all the fluorescent images were taken with fixed exposure settings: 4 s for RFP imaging and 12 s for GFP imaging at 40× magnification; 1 s for RFP imaging and 3 s for GFP imaging at 100× magnification; and 0.5 s for RFP imaging and 2 s for GFP imaging at 200× magnification. In some instances, Hoechst 33342 (1 µg/mL) was added to the culture to photograph the cell nucleus with a DAPI filter at 100× magnification for 30 s. The Layer Style Blending Option of Photoshop CS4 (Adobe Systems, San Jose, CA, USA) was used to demonstrate the localization of green or red fluorescence in cultured cells.

## 3. Results

LNCaP and HPE-15 cells share behaviors that are relevant to the early stages of human prostate cancer. As previously mentioned [19,20,22], these cells do not form tumors in immunocompromised mice. However, when co-cultured with bystander cells or other cancer cells, LNCaP and HPE-15 cells become heterogeneous. Some of these cells acquire tumorigenic properties and display signs of metastasis in internal organs. We have identified fusion between cancer cells and bystander cells as a mechanism for heterogeneity progression [19,20]. Therefore, we conducted a study to investigate the effects of fusion on LNCaP^RFP^ and HPE-15^RFP^ cells when interacting with bystander stromal cells.

### 3.1. Spontaneous Transformation of Prostate Mesenchymal Stromal Cells

We used a cancer–stromal cell co-culture to model the interaction between LNCaP^RFP^ or HPE-15^RFP^ cells and six pair-matched prostate mesenchymal stromal cell lines derived from three prostate cancer patients [23]. As a control, we used PrSC cells from a healthy donor. All seven stromal cell lines used in this study were between passages 31 and 35 since their original cloning. Our findings showed that approximately 20% of the LNCaP^RFP^ cells formed fusion hybrids with stromal cells within a week of co-culture, which was consistent with our previous reports [19].

In contrast, HPE-15^RFP^ cells exhibited much higher fusogenicity. In fact, after just 24 h of co-culture, all individual HPE-15^RFP^ cells had fused with the stromal cells. These fusion hybrids appeared as large stromal cells emitting RFP fluorescence, mostly with two nuclei, which aligns with our previous descriptions. Unlike LNCaP^RFP^ cells, HPE-15^RFP^ cells exhibited a uniform epithelial cell morphology in cobblestone-like growth [22]. Using this feature to track cellular heterogeneity, we focused on tracking morphologic changes to monitor heterogeneity progression in HPE-15^RFP^ cells.

Interestingly, after keeping the co-cultures for eight weeks to track the fate of the fusion hybrids, only the co-culture with HPS-11 cells resulted in hybrid-derived cells. These cells assumed a small stromal cell shape but emitted red fluorescence. At the end of the eight-week observation period, the co-culture with HPS-11 cells showed colonies expressing RFP in various morphologies, indicating that they were derivative clones from the fusion of cancer and stromal cells. In contrast, we found very few derivative clones in the HPS-10, HPS-12, HPS-13, HPS-14, HPS-15, or PrSC co-cultures. In these cases, the large hybrids with mesenchymal stromal morphology and RFP fluorescence remained senescent throughout the entire eight weeks, showing no signs of cell division.

After repeated studies, we concluded that, while all prostate stromal cells were highly susceptible to fusion with HPE-15^RFP^ prostate cancer cells, the HPS-11 cells were unique in their ability to facilitate the proliferation of the hybrids into derivative clones.

The HPS-11 cell line was isolated as a cancer-associated mesenchymal stromal cell line, paired with HPS-10, which was from the normal zone of the same tumor resection [23]. Microscopic inspection revealed that the HPS-11 population between passages 31 and 35 consisted of both large and small stromal cells. This distinction became apparent when HPS-11 was replated at low density. While most of the population consisted of large, elongated cells with slow growth, a few small, spindle-shaped cells with accelerated proliferation were also present, forming colonies within 7 days (Figure 1A,B). In co-culture experiments, it was observed that hybrid colonies were mostly derived from the fusion of HPE-15^RFP^ with these small stromal cells.

Historically, HPS-11 cells were isolated through cloning, and the small, spindle-shaped cells with a fast growth rate were not identifiable during the first 30 passages [23]. These cells most likely emerged through spontaneous transformation during extended culture [24,25], although no transformation was observed in the other six stromal cell lines after similar passages. We named these transformed cells HPS-11t.

To verify the origin of the transformed cells, we isolated six HPS-11t clones from the low density HPS-11 culture. Another six clones were isolated from the limiting dilution cloning of the GFP-tagged HPS-11 cells and named HPS-11t^GFP^ clones. The HPS-11 origin of these clones was confirmed with STR profiles identical to the original HPS-11. The HPS-11t clones exhibited significantly faster growth rates compared to the original HPS-11 cells and even faster than the HPE-15 cells (Figure 1C). Since these cells seemed to be responsible for the increased derivative hybrid colony formation, our focus in the subsequent study was to assess the role of HPS-11t cells.

### 3.2. Survival and Proliferation of Fusion Hybrids from the Co-Culture with HPS-11t Cells

We conducted co-cultures using HPE-15^RFP^ cells with either HPS-11t or HPS-11t^GFP^ clones to observe what happened to cancer–stromal fusion hybrids. Consistent findings were observed in all co-cultures, so we will present representative outcomes from the co-culture of HPE-15^RFP^ with HPS-11t^GFP^-1. Both cell lines were obtained through limiting dilution cloning. The HPE-15^RFP^ cells had a typical epithelial appearance (Figure 2A), while the HPS-11t^GFP^-1 clone displayed a small spindle shape (Figure 2B). With distinct fluorescence proteins, we were able to track the cancer–stromal interaction in real-time [19].

Real-time fluorescence microscopic tracking revealed rapid fusion between cancer and stromal cells, as cells with both red and green fluorescence could be seen as early as 12 h after the co-culture began (Figure 3A). At this time, most of the hybrids appeared as small and rounded cells. Subsequently, hybrids with typical stromal morphology started to appear and became more frequent around 7 days of the co-culture (Figure 3B). After that, the co-culture became more complex, with clusters of dual fluorescent cells in various shapes emerging in the epithelial HPE-15^RFP^ and stromal HPE-11t^GFP^-1 co-culture. We continuously tracked this and found that it was caused by fusion hybrids that began to divide. Although the co-culture often contained dead cells with dual fluorescence, indicating hybrid death [19,20], the division of fusion hybrids in the HPE-15^RFP^ and HPS-11t^GFP^-1 co-culture was equally effective. Within 21 days, the entire co-culture was scattered with clusters of fusion hybrid derivative cells (Figure 4A,B).

The derivative cells obtained from the fusion hybrids exhibited diverse morphologies and distinct growth behaviors. Some cells had an epithelial morphology, arranged in a cobble stone pattern, and had a slow growth rate (Figure 4A). Other cells took on a stromal cell shape, forming colonies that grew rapidly and were scattered (Figure 4B). Additionally, there were derivative cells that had transitional shapes, transitioning from epithelial to stromal cells, and displayed varying growth rates. The fusion of cancer and stromal cells, therefore, could give rise to hybrid progeny with diverse morphologies and growth behaviors.

The widespread proliferation of hybrids was confirmed through continued passaging with low density replating. Daily inspection revealed two distinct behaviors of the fusion hybrids. About 40% of the hybrids displayed limited cell division, undergoing a single or no division following replating. The hybrids with no division were generally large in size, with large singular or plural nuclei indicating growth arrest in the heterokaryon or synkaryon state. Similarly, the hybrids with a singular division were growth arrested as well. These hybrids remained in the two-cell state until the end of an entire 28-day culture period without signs of additional division (Figure 5).

In contrast, about 60% of the fusion hybrids displayed the capability of repeated division, forming colonies individually with morphologies and growth behaviors that diversified from each other (Figure 6).

These results demonstrated that, while some lacked replication capability, a substantial number of fusion hybrids from the HPE-15^RFP^ and HPS-11t^GFP^-1 co-cultures could proliferate to form a derivative population with diversified morphology and growth behavior. Recognizing the implications for clinical cancer heterogeneity progression, we focused on this phenomenon by tracking the fate of fusion hybrids with unlimited proliferation.

### 3.3. Heterogeneity of the Fusion Hybrid Derivative Clones

Our co-culture of fluorescently tagged cells provides a unique opportunity to study the effects of cancer–bystander cell interaction. Our objective was to isolate derivative clones in order to measure heterogeneity among the individual fusion hybrids.

Derivative clones were isolated from low-density plating for further examination. A total of 192 dual red- and green-fluorescent clones (referred to as RG-clones) were picked and cultured for 30 continuous passages from a single HPE-15^RFP^ and HPS-11t^GFP^-1 co-culture. These clones exhibited epithelial, stromal, or transitional morphologies to varying degrees. The analysis of this series of studies revealed a marked difference in growth rates. Clones with stromal or transitional shape displayed a stable morphology and exhibited fast proliferation. Epithelial clones in early passages showed noticeable instability, with individual cells in each clone exhibiting diversification in terms of cell size, nucleus sizes or numbers, and green or red fluorescence intensity (Figure 7). Furthermore, the growth of epithelial clones was much slower compared to stromal clones. However, this instability was attenuated over time, as cells in epithelial clones became smaller and polykaryons became less frequent during successive passaging. After 10 passages, cells in the epithelial clones became predominantly uniform and all these clones survived the continuous passaging. Both phase contrast microscopic examination and Hoechst 33342 staining indicated the presence of a singular nucleus in most cells. These results demonstrated that a substantial number of the HPE-15^RFP^ and HPS-11t^GFP^-1 fusion hybrids were able to perpetuate and propagate successfully.

At passage 30, the clones were categorized into three groups based on their morphologic attributes: epithelial-like, transitional, and stromal. For further characterization, four clones were randomly selected from the epithelial group (RG-21, 86, 90, and 144, Figure 8A), and another four were chosen from the stromal group (RG-9, 46, 136, and 183, Figure 8B). All these randomly selected clones appeared to be unique, displaying distinctive phenotypic attributes and growth behaviors. This suggested a significant inter-clonal heterogeneity. These eight clones were used in the following studies.

To determine clonality and clonal stability, the eight clones were cultured continuously until passage 60, following the previously described method [19]. The stromal clones took approximately eight months to culture from passage 30 to passage 60, using a replating ratio of 1:5. The epithelial clones, on the other hand, took about twelve months to reach the same passage. Throughout this culturing process, no changes in morphology or growth behavior were observed.

STR genotyping analyses were conducted to examine the hybridization between the HPE-15^RFP^ and HPS-11t^GFP^-1 cells (Table 1). Results showed multiple allelism in all the loci examined and in all eight randomly selected clones, indicating the presence of genomic materials from both parental cells. Meanwhile, there was extensive genomic heterogeneity, with individual clones displaying different combinations of allelic polymorphism. No clones were found to have an identical genotyping profile. This heterogeneity was so thorough that no correlation could be established between the genotypic profile and the phenotypic traits of the clones.

To evaluate the inheritance of fusogenicity, the eight clones were subjected to another round of co-culture. Once again, rapid and effective fusion was observed with the parental HPS-11t^GFP^-1 stromal cells. However, the fate of the fusion hybrids was not tracked, and the fusogenicity of the eight clones was not compared to their parental HPE-15^RFP^ cancer cells.

As a proof-of-concept study to assess tumor formation potential, we tested two epithelial clones (RLG-21 and RLG-86) for xenograft tumor formation. The parental HPE-15^RFP^ cells were non-tumorigenic [22]. However, both derivative clones acquired tumorigenicity (Figure 9). Interestingly, while the parental HPS-11 cells were non-tumorigenic [23], the HPS-11t^GFP^-1 clone formed subcutaneous tumors. Whether the tumorigenicity in the derivative clones is inherited from the transformed stromal cells or acquired through the process of cell fusion remains to be determined.

## 4. Discussion

This study investigated the acquisition of heterogeneity in HPE-15^RFP^ prostate tumor cells through fusion with mesenchymal stem/stromal cells. While fusion hybrids were common in the co-culture with all stromal cells evaluated, only the hybrids with transformed stromal cells proliferated into derivative clones. These clones displayed significant inter-clonal heterogeneities and retained the ability to produce more hybrid progenies in further co-culture. The observed genomic heterogeneity and cellular complexity were more extensive than our previous estimates, supporting the “dark matter” hypothesis [26]. This hypothesis suggests that cell fusion in clinical tumors is frequent and dynamic but difficult to detect or quantify due to isogenicity between the involved cells. As cancer interacts with bystander cells in the tumor microenvironment and co-evolves with stromal cells [21], the progression of heterogeneity induced by cell fusion must be a continuous process throughout the entire history of disease progression. The implications of our study will be carefully considered.

### 4.1. Transformed Bystander Cells Are More Permissive to Heterogeneity Progression

By tracking the formation and fate of fusion hybrids, we have established an experimental system to investigate the natural history of cancer–bystander cell fusion in real time. We have found that fusion with mesenchymal stem/stromal cells can have opposing effects [19,20]. It can either prevent the progression of heterogeneity by trapping cancer cells in the heterokaryon or synkaryon state or facilitate progression if the fusion hybrids manage to escape the trap and re-enter the cell cycle. Mesenchymal stem/stromal cells are known to be prone to spontaneous transformation, especially during long-term expansion [24,25]. In our comparative analysis, we have discovered that transformed stromal cells have a significantly increased capacity to facilitate the re-entry of fusion hybrids into the cell cycle. In clinical tumors, chronic interaction with tumor cells likely leads to the transformation of bystander mesenchymal stromal cells, which can be recruited through fusion with tumor cells and reprogrammed into heterogeneous constituents of the tumor cell compartment. This study identifies abnormalities in bystander cells as contributing factors to the progression of heterogeneity.

How did the transformed stromal cells facilitate the proliferation of fusion hybrids? Although the underlying mechanism remains to be elucidated, it is conceivable that the accelerated growth rate in transformed stromal cells is compatible with hybrid cell division. Transformed HPS-11t^GFP^-1 cells exhibited enhanced proliferative activity, with a much faster growth rate than the parental HPS-11 stromal cells. Coincidentally, the transformed cells showed a remarkable capacity to facilitate the proliferation of fusion hybrids into derivative clones. Since the parental stromal cells grow much more slowly than the HPE-15^RFP^ cells, their fusion hybrids are controlled by two different cell cycle timings. The division of the hybrids becomes challenging due to dysregulated cytokinesis or karyokinesis, leading to growth arrest and mitotic catastrophe [27]. However, in the co-culture with transformed stromal cells, both the cancer and stromal cells were actively proliferating, despite their cycles not being synchronized. As a result, the division of their fusion hybrids is now controlled by similar or compatible cytokinesis and karyokinesis rhythms, providing an opportunity for the hybrids to re-enter the cell cycle. The growth rate of bystanders is a critical determinant of the fate of fusion hybrids.

This proposition is supported by our unpublished observations from previous co-culture studies. While LNCaP prostate cancer cells could fuse with slow-growing bone marrow- or prostate-derived mesenchymal stromal cells, the fusion hybrids had limited chances to proliferate into derivative clones [19]. In contrast, many derivative clones were observed from the LNCaP co-culture with fast-growing virally transformed HS-5 and HS-27a bone marrow-derived mesenchymal stromal cells [28]. Further investigation is necessary to uncover the mechanism by which transformed stromal cells facilitate the proliferation of fusion hybrids.

### 4.2. Individual Fusion Events Are the Modular Units of Tumor Cell Heterogeneity

Our studies have revealed several clues suggesting that individual cell fusion events have the potential to introduce heterogeneity directly into the tumor population. The HPE-15 is a non-tumorigenic prostate tumor cell line with a karyotype that is more similar to virally transformed prostate epithelial cells than commonly used prostate cancer cell lines. It is intriguing that such a cell line can acquire tumorigenicity simply through cell–cell interaction [22]. In addition to this study, our unpublished works have detected HPE-15^RFP^ fusion with other prostate cancer cells, in agreement with findings that malignancy can spread among tumor cells [29,30]. HPE-15 cells can also fuse with each other. These findings may explain the fact that, even in a newly established cancer cell line like HPE-15, individual cells have heterogeneous genomic makeups [22], while the karyotype description for a given cancer cell line always includes structural and numerical variations [31]. Cell fusion is a straightforward way for tumor cells to acquire heterogeneity.

Each fusion event has the potential to introduce unique heterogeneity to the tumor cell population. In the current study, one round of HPE-15^RFP^ co-culture with transformed stromal cells yielded abundant fusion hybrid progenies with extensive genomic hybridization and inter-clonal heterogeneity. All eight randomly selected clones were found to have multiple allelism, which is consistent with hyperdiploidy in clinical tumor cells. Although STR analysis cannot determine copy numbers, the presence of multiple allelism suggests a minimum doubling of the genome. About 60% of the fusion hybrids were able to undergo repeated cell division, which is a critical requirement for somatic mutation [32]. Through cell division, heterokaryons and synkaryons undergo extensive genomic reorganization to reduce the chromosome number [33], likely using a meiosis-like process of diploidization [34], during which genetic recombination occurs at random locations [35]. The inherent aneuploidy of the parental cancer cell, combined with chromosomal mismatches in the hybrid, makes the meiosis-like process incomplete and unstable, resulting in hybrid progenies with dynamic aneuploidy, perceived as genomic instability [20]. At the same time, the spontaneity of fusogenicity inherited by hybrid progenies may drive cell fusion in the tumor microenvironment to yield more heterokaryons, perpetuating the dynamic aneuploidy throughout the clinical history of disease progression.

In clinical tumors, cancer cells may fuse with neighboring bystander cells. In the simplest scenario, somatic fusion involves a single cancer cell and a specific bystander cell. Although the cancer and bystander cells involved are isogenic, their resulting fusion hybrid becomes heterogeneous. This is because each fusion represents an isolated event of genomic hybridization. The subsequent genomic reorganization, diploidization, and genetic recombination take place within the hybrid, separate from any neighboring cells or fusion hybrids. Since recombination events are randomly distributed [35], each fusion event creates a derivative clone with a mutated genomic makeup that is unique and distinct from other cancer–bystander cell fusion hybrids in the same tumor. The findings of this study suggest that individual fusion events may potentially contribute to the progression of tumor cell heterogeneity.

### 4.3. The Dynamic Heterogeneity Confounds Tumor Marker Identification

The aberrant reactivation of gene expression is a well-known phenomenon related to tumors and is a clear indication of dysregulated genetic control. Eukaryotic cells, including tumor cells, appear to be highly tolerant of reactivated gene expression. Except for a few lethality-related genes [36] or those involved in synthetic lethality [37], the reactivation and even overexpression of most protein-encoding genes do not affect the survival of tumor cells. Interestingly, although many non-housekeeping genes are found to be abnormally expressed in tumors, only a few can be defined as tumor markers and even fewer as companion biomarkers. This is because the candidate genes always fail to reach a consensus pattern of temporal or spatial expression in the majority of tumor cells [38]. Few candidates can be scored with a prevalent expression or a close correlation to the progression of the disease through histopathologic quantification. The reactivation and overexpression of genes in cancers seem to be rather erratic or random [39].

The results of this study suggest that heterogeneity is a result of random events at the level of individual tumor cells. The randomness and individuality make it impossible to profile tumor cell heterogeneity with a consensus pattern. We propose that the spontaneity of fusogenicity in cancer–bystander cell fusion, combined with the randomness of subsequent genetic recombination, may explain the confounding issue in tumor marker identification. Gene expression and protein production are controlled by various biochemical mechanisms, which cause expression noises and fluctuations due to the stochastic nature of biochemical reactions with finite substrate molecules [40,41,42]. Even in normal tissues, the expression of a gene among individual cells is often heterogeneous. Gene expression and protein levels can be transiently increased or decreased by epigenetic regulation and other non-mutational mechanisms [43]. In fusion hybrid derivative cells, the expression heterogeneity could only be strengthened, while the complete genomic heterogeneity and prevalent multiple allelism may cause drastic changes in gene expression. Most importantly, although a given gene is regulated by numerous factors at different levels, the output of its expression is inherently binary, limited to an “on” or “off” mode in an all-or-none fashion [44]. Instead of widespread expression, a completely dysregulated marker gene would be “on” in 50% of the tumor cell population. A combinatorial marker analysis will only decrease the positivity score because, in cancer cells, each marker expression is abnormally regulated by independent mechanisms, following the multiplication rule of probability.

### 4.4. Limitations of the Study

This study has suggested that, compared to their parental cells, transformed prostate stromal cells are more effective in promoting the proliferation of cancer–stromal fusion hybrid progenies in the tumor microenvironment. These hybrid progenies exhibit a high level of heterogeneity in terms of genotype and phenotype. Heterogeneity has the potential to evolve if cancer cells continue to fuse. Heterogeneous hybrids may acquire additional cancer hallmarks [45] and be positively selected. Further research is needed to fully understand the clinical significance of cancer–bystander cell fusion, while the treatment of tumor progression and metastasis through inhibiting spontaneous cancer–bystander cell fusion remains to be tested.

One perplexing observation in this study is the choice between growth arrest and proliferation. While transformed stromal cells are more supportive than parental stromal cells, approximately 40% of the fusion hybrids displayed limited division, while the remaining 60% exhibited unrestricted proliferation. Comparative studies are needed to explore the underlying mechanism and identify the determinants that are essential for tumor cell survival and proliferation.

Fusion with HPS-11t^GFP^-1 cells resulted in a significant number of derivative clones, whereas fusion with non-transformed prostate stromal cells did not produce any hybrid progenies. This contrasts with previous studies where fusion between RFP-tagged LNCaP cells and non-transformed stromal cells did generate hybrid derivative clones, although with limited success. Both LNCaP and HPE-15 are non-tumorigenic prostate cancer cell lines. The cell fusion mechanism may be constitutively expressed in HPE-15^RFP^ cells, which facilitates rapid (less than 24 h of co-culture) and complete (100% of the HPE-15^RFP^ cells) fusion with stromal cells. On the other hand, the slow (over 48 h of co-culture) and incomplete (around 20% of the cells) fusion observed with RFP-tagged LNCaP cells indicates an induced mechanism. Comparative analyses are required to determine the mechanism responsible for cancer–stromal cell fusion.

While a large number of derivative hybrid clones were obtained from the fusion between HPE-15^RFP^ and HPS-1t^GFP^-1 cells, the heterogeneity of these clones has only been characterized in terms of clonal morphology, growth behavior, and genotyping. The extent of phenotypic heterogeneity among the clones needs to be evaluated. Representative clones should be tested for their ability to form xenograft tumors and their sensitivity to anti-tumor drug treatments. Comparative analyses should be conducted to demonstrate the impact of prostate cancer cell fusion with transformed stromal cells on the progression and metastasis of the disease.

It is important to note that this work is mostly observational and qualitative, as few quantitative parameters on the cancer–bystander cell fusion were analyzed. Future studies will aim to perform a molecular and pathophysiological characterization of fusion-derived cancer cells.

## 5. Conclusions

Tumor cells acquire heterogeneity through interaction with the tumor microenvironment, which has a complex milieu with various bystander cells, while the interaction takes place in three-dimension with dynamic transitions. To assess the role of cancer–bystander cell fusion in tumor cell heterogeneity, we used a cancer–stromal co-culture model to identify whether transformed stromal cells could facilitate the proliferation of fusion hybrids, leading to complete tumor cell heterogeneity. A comparative study of the transformed stromal cells will identify the underlying mechanism. The cancer–bystander co-culture model is a powerful tool for mechanistic investigation in tumor cell heterogeneity progression.

## Figures and Tables

**Figure 1 cancers-16-00951-f001:**
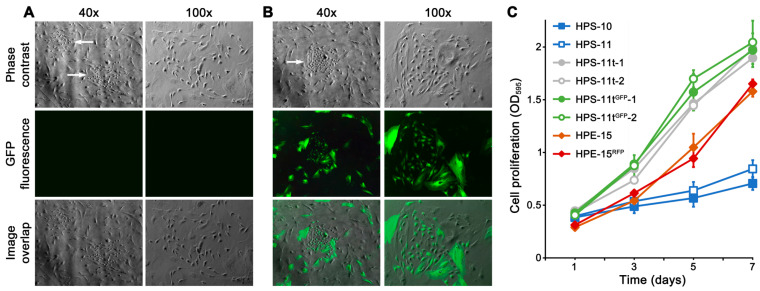
Spontaneous transformation of HPS-11 prostate mesenchymal stromal cells. Growth behavior of the clone is shown at 40× and 100× magnifications. (**A**) HPS-11 cells at passage 35 were subjected to low density plating (1:20). Transformed cells, the HPS-11t cells, formed clones (arrows) after 7 days of culture. (**B**) HPS-11 cells at passage 35 infected with TurboGFP lentiviral particles were found to have a similar spontaneous transformation (arrow). (**C**) Cells used in this study were tested for their growth rates by crystal violet staining. Two HPS-11t and two HPS-11t^GFP^ clones were included for comparison. HPS-10 and HPS-11 at passage 16 were used for comparison. Each data point represents the average of a triplicate assay ± standard deviation.

**Figure 2 cancers-16-00951-f002:**
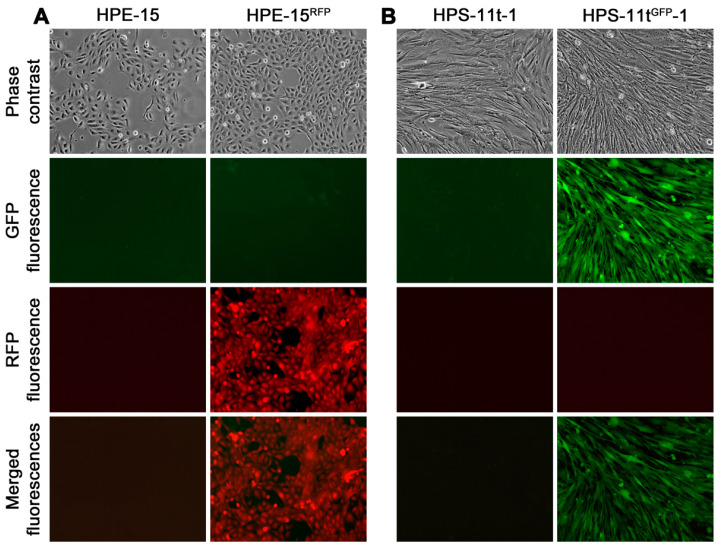
Distinctive morphology and growth behavior of cells used in co-culture. (**A**) The epithelial morphology of HPE-15 cells and the HPE-15^RFP^ clone. (**B**) The stromal shape of the HPS-11t-1 and HPS-11t^GFP^-1 clones. All images are at 100× magnification.

**Figure 3 cancers-16-00951-f003:**
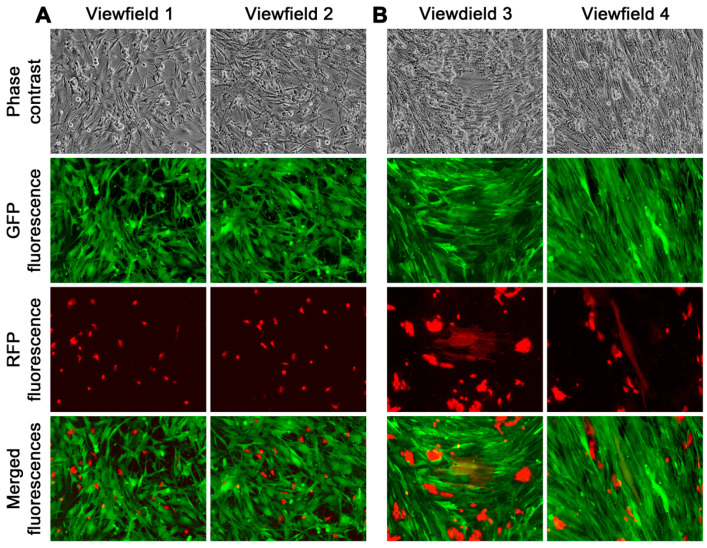
Effective cancer–stromal cell fusion. (**A**) The images were taken 12 h after the introduction of HPE-15^RFP^ cells to the HPS-11t^GFP^-1 monolayer. Cancer–stromal cell fusion could already be seen at this time, as indicated by the overlapped fluorescence images (merged fluorescence). (**B**) Seven days into the co-culture, frequent cancer–stromal cell fusion could be seen. All images are shown at 100× magnification.

**Figure 4 cancers-16-00951-f004:**
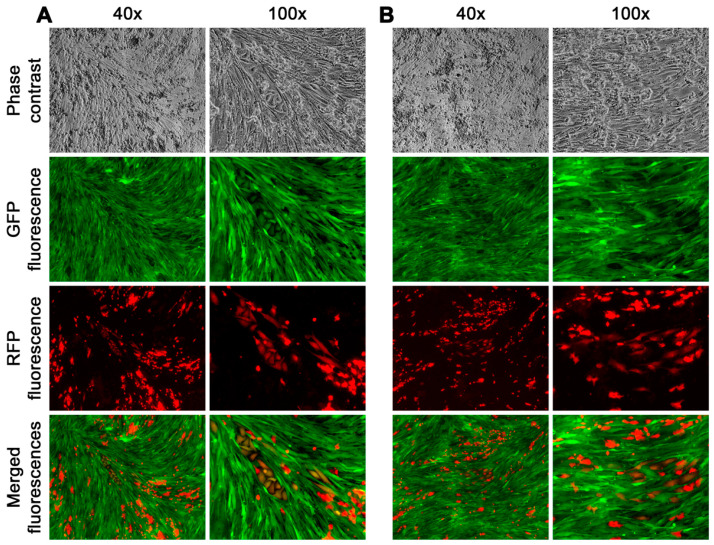
Efficient survival and proliferation of the fusion hybrids during co-culture. Representative view fields of a co-culture are shown. (**A**) The images were taken at 21 days into the co-culture. A view field encompassing a derivative clone with an epithelial morphology is shown at 40× and 100× magnifications. (**B**) Another view field shows a clone with stromal cell shape.

**Figure 5 cancers-16-00951-f005:**
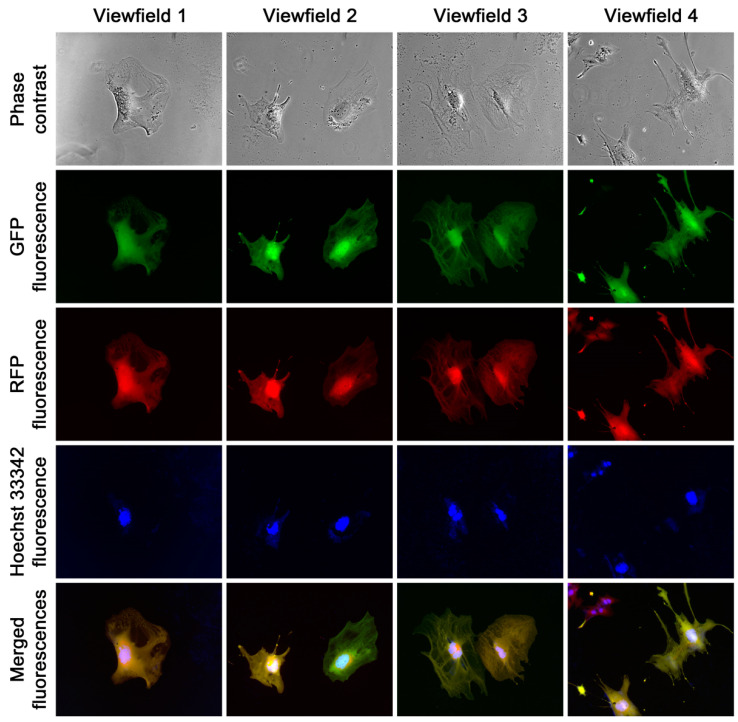
Survival without proliferation was observed in approximately 40% of the fusion hybrids. In this presentation, four representative view fields were taken from a 21-day culture following low-density replating. Blue fluorescence detected cell nucleus stained by Hoechst 33342. All images are shown at 100× magnification.

**Figure 6 cancers-16-00951-f006:**
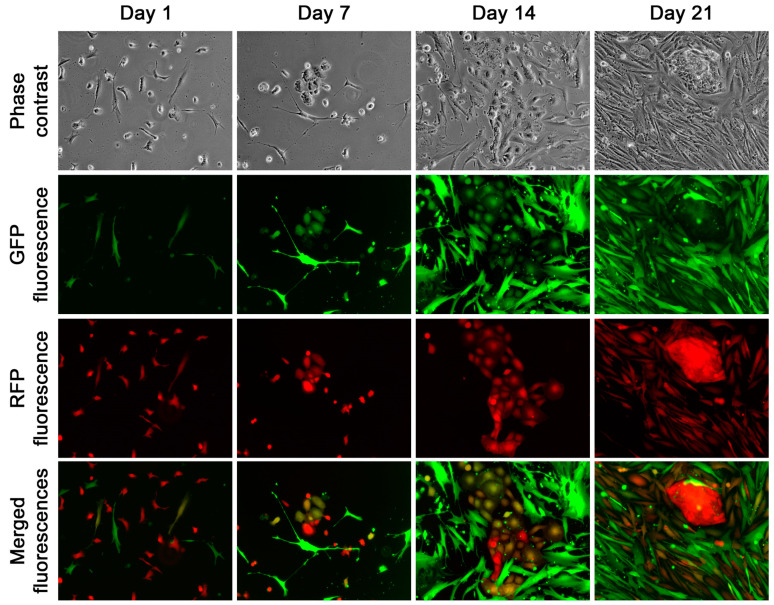
Survival and proliferation observed in the other approximately 60% of the fusion hybrids. In this presentation, representative view fields were taken every 7 days during a 21-day culture following low-density replating. All images are shown at 100× magnification.

**Figure 7 cancers-16-00951-f007:**
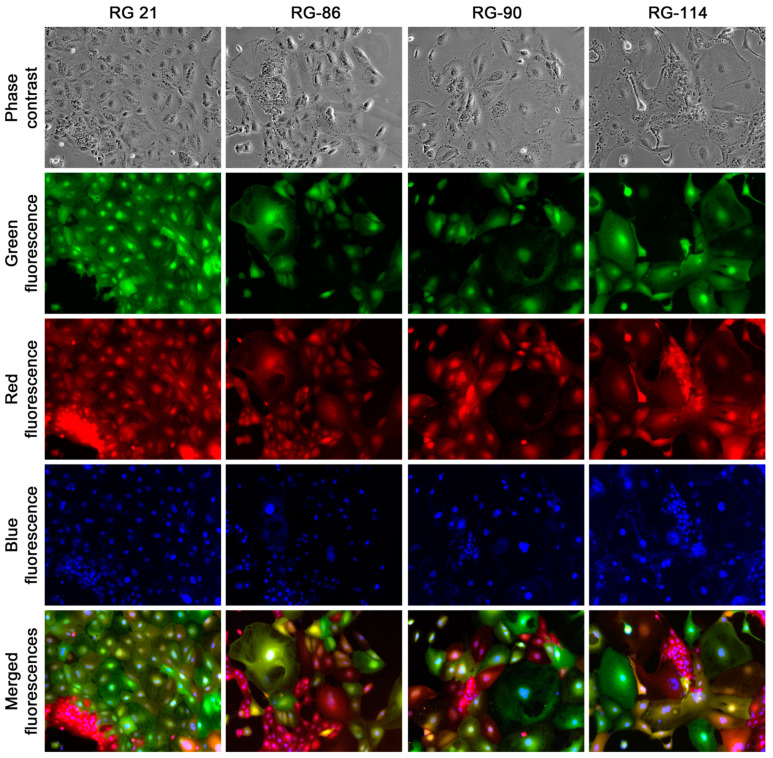
Instability of the epithelial clones derived from the cancer–stromal cell fusion hybrids. In this presentation, intra-clonal instabilities in four epithelial clones were shown at passage 3. Blue fluorescence detected cell nucleus stained by Hoechst 33342. All images are shown at 100× magnification. With continuous passaging, cells in all four clones eventually became uniform, and the clones were cultured beyond passage 30.

**Figure 8 cancers-16-00951-f008:**
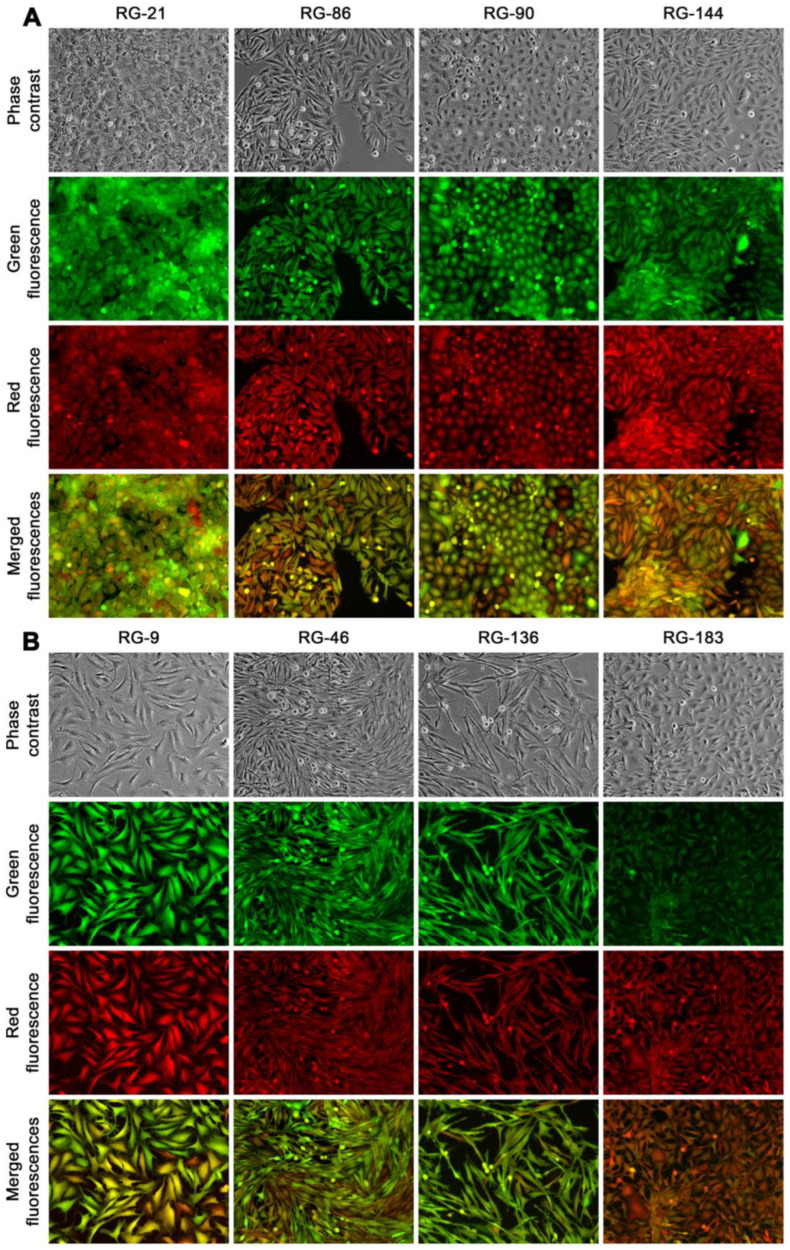
Derivative clones from cancer–stromal cell fusion. In this presentation, eight clones at passage 62 were chosen to demonstrate inter-clonal heterogeneity in morphologies and growth behaviors. (**A**) Four clones with epithelial morphologies are shown. Note the unique features of overlapping growth (RG-21), signs of epithelial to mesenchymal transition (RG-86), cobblestone arrangement (RG-90), and compacted growth (RG-144). (**B**) Four clones with mesenchymal stromal morphologies are shown. Note the unique features of large cells in scattered growth (RG-9), small cells in longitudinal growth (RG-46), long and thin cells in parallel growth (RG-136), and small cells with dendritic extensions (RG-183). All images are shown at 100× magnification.

**Figure 9 cancers-16-00951-f009:**
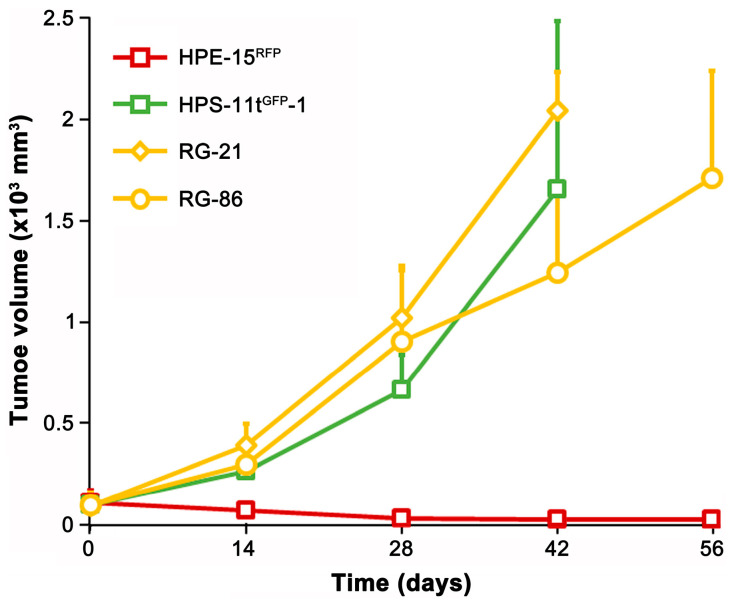
Acquired tumorigenicity from cancer–stromal fusion. Derivative clones RG-21 and RG-86 were assessed for xenograft tumor formation in nude mice (*n* = 5). Tumor volume of 1.5 cm^3^ was used as humane endpoint. Each data point represents the average ± standard deviation.

**Table 1 cancers-16-00951-t001:** Genomic heterogeneities of the fusion hybrid derivative clones detected by STR analysis ^a^.

STR locus	HPE-15 ^b^	HPE-15^RFP^	RG-9	RG-21	RG-46	RG-86	RG-90	RG-136	RG-144	RG-183	HPS-11t^GFP^-1	HPS-11
D3S1358	15, 17	15, 17	14, 15, 16	14, 15, 16, 17	14, 15, 16, 17	14, 15	14, 15, 16, 17	14, 15	14, 15, 16, 17	14, 15, 16	14, 16	14, 16
FGA	19, 25	19, 25	19, 24, 25	24, 25	19, 24, 25	19, 24, 25	24, 25	19, 24, 25	19, 24, 25	19, 24	24	24
D5S818	10, 13	10, 13	11, 13	10, 11, 13	10, 11, 13	10, 11, 13	10, 11, 13	10, 11, 13	10, 11, 13	10, 11, 13	11, 13	11, 13
CSF1PO	10, 11	10, 11	9, 10, 13	9, 10, 11, 13	9, 10, 11, 13	9, 10, 11, 13	9, 10, 11, 13	9, 10, 11, 13	9, 10, 11	9, 11, 13	9, 13	9, 13
D7S820	10, 13	10, 13	8, 10, 12, 13	10, 12, 13	8, 10, 12, 13	8, 10, 12, 13	8, 10, 12, 13	8, 10, 12, 13	8, 10, 12, 13	8, 10, 12, 13	8, 12	8, 12
D8S1179	14, 15	14, 15	11, 14, 15	11, 15	11, 14, 15	11, 14, 15	11, 14, 15	11, 14, 15	11, 15	11, 15	11, 15	11, 15
TH01	9.3	9.3	7, 9, 9.3	9, 9.3	9, 9.3	7, 9, 9.3	7, 9, 9.3	7, 9, 9.3	9, 9.3	7, 9, 9.3	7, 9	7, 9
vWA	14, 17	14, 17	14, 16, 17, 19	14, 16, 19	16, 17, 19	14, 16, 17, 19	14, 16, 17, 19	14, 17, 19	14, 17, 19	14, 16, 17, 19	16, 19	16, 19
D13S317	11	11	8, 11	8, 11	8, 11	8, 11	8, 11	8, 11	8, 11	8, 11	8	8
Penta_E	11, 12	11, 12	7, 11, 12, 14	7, 12, 14	7, 11, 12, 14	7, 12, 14	7, 11, 12, 14	7, 11, 14	7, 12, 14	7, 11, 14	7, 14	7, 14
D16S539	11, 12	11, 12	9, 11, 12	9, 11	9, 11, 12	9, 11, 12	9, 11, 12	9, 11, 12	9, 11, 12	9, 11, 12	9	9
D18S51	14	14	14, 16	14, 16	14, 16	14, 16	14, 16	14, 16	14, 16	14, 16	16	16
D19S433	14, 15	14, 15	13, 14, 15	13, 15	13, 14, 15	13, 14, 15	13, 14, 15	13, 14, 15	13, 14, 15	13, 14, 15	13, 15	13, 15
Penta_D	12, 13	12, 13	10, 13	10	10, 12, 13	10, 13	10	10, 13	10, 12, 13	10 13	10	10
D21S11	28, 30	28, 30	28, 29	28, 29, 30	28, 29, 30	28, 29	29	28, 29	28, 29, 30	28, 29	29	29

^a.^ Only the STR analysis results that are informative to the current study are shown to protect identity of original donors. ^b.^ STR counts from HPE-15^RFP^ cells are indicated in red, and STR counts from HPS-11t^GFP^-1 cells are in green. For the hybrid clones, STR counts that were not informative for this analysis are indicated in black.

## Data Availability

Data that support the findings of this study are available from the corresponding author upon request.

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
