# Peer review of "Spontaneous Fusion with Transformed Mesenchymal Stromal Cells Results in Complete Heterogeneity in Prostate Cancer Cells"

_cancers, 2024, doi:10.3390/cancers16050951_

Round 1
Reviewer 1 Report
Comments and Suggestions for Authors
1. In figures 1-8, the labeled proteins in red and green should be specified. "red" and "green" is not a proper specification of the fluorescent labeling.
2. Recent finding indicating that different proteins cause post-translational modification of NuMA in healthy and malignant epithelial cells can be used for selective eradication of malignant epithelial cells. Drug Dis. Today 27, 2022
Author Response
The following is a point-by-point response to reviewers’ comments.
Reviewer 1.
- In figures 1-8, the labeled proteins in red and green should be specified. “red” and “green” is not a proper specification of the fluorescent labeling.
Our response: We thank the reviewer for this suggestion. In the revised Figures, the “Red fluorescence”, “Green fluorescence”, and “Blue fluorescence” labels have been changed to “RFP fluorescence”, “GFP fluorescence”, and “Hoechst 33342 fluorescence”, respectively.
In addition, font size of the labels was enlarged and data labels in Figure 1 were changed to be consistent with the main text.
- Recent finding indicating that different proteins cause post-translational modification of NuMA in healthy and malignant epithelial cells can be used for selective eradication of malignant epithelial cells. Drug Dis. Today 27, 2022.
Our response: Following the Reviewer’s introduction, we studied this inspirational review article, which showed that the nuclear mitotic apparatus (NuMA) protein was a critical modulator of spindle formation in mitosis, as inhibition of post-translational modification of this protein resulted in mitotic catastrophe, the type of cell death observed in our study.
We are indebted to the Reviewer’s constructive suggestion. In future studies to characterize the 192 individual clones derived from fusion hybrids, we will examine the involvement of NuMA in cancer-bystander cell fusion to decipher the mechanism of mitotic catastrophe in the fusion.

Reviewer 2 Report
Comments and Suggestions for Authors
This is a well conducted study regarding the prostate stromal cell transformation, cell fusion which contribute the heterogenicity of prostate cancer and treatment resistance.
Some minor comments:
1. are these findings of current study can be found in " human" prostate cancer ?, like the "transormated stromal cell" also showed in human advanced or CRPC status of prostate cancer?
2: how about the tumor growth rate and ability of metastasis of theses derivated clones of cancer-stromal cell? also, did these colony AR dependent or AR indepedentent?
3: Tumor microenvironment is not only epithelial and stromal cells, there other cells like immune cells, macrophage, vessels cell should also discusses is the limitation part
Comments on the Quality of English Language
The English is good
Author Response
The following is a point-by-point response to reviewers’ comments.
Reviewer 2.
- are these findings of current study can be found in “ human” prostate cancer ?, like the “transormated stromal cell” also showed in human advanced or CRPC status of prostate cancer?
Our response: We thank the Reviewer for this highly relevant issue to clinical prostate cancer. It is technically difficult to study stromal cell transformation in clinical tumor specimens. This is due to the following technical issues.
1) In clinical cancers, the stromal origin can not be determined. Current histopathology is unable to determine whether a “transformed stromal cell” is indeed derived from stromal cell, or from a cancer cell instead. At the present, cancer research field is centered around “cancer cells”. Consequently, appearance of “transformed stromal cells” would be considered as polymorphic cancer cells. In clinical tumor specimens, for example, any potential “transformed stromal cells” would be designated as “reactive stroma”, “epithelial-mesenchymal transition (EMT)”, or “lineage plasticity of cancer stem cells”.
2) One of the surest methods to identify cell origin is tracking the cell’s genetic fingerprints. Within a patient tumor specimen, however, cancer and stromal cells share identical genetic fingerprints. It is technically challenging to use genetic fingerprinting technologies to tell whether a heterogeneous cell in the tumor microenvironment is originated from a cancer or a bystander stromal cell.
3) Despite cancer cell is currently thought to be the sole player in cancer progression and metastasis, transformed cells that are originated from the stromal compartment can be detected experimentally, in xenograft tumors, since human cancer cells can be easily distinguished from stromal cells of the mouse host. In the revised manuscript, we emphasized this fact in the “Introduction” section (lines 72-76). Relevant references are cited.
4) For experimental purpose, fluorescent protein technology is another way to determine “stromal cell origin” of transformed cells. As shown in the current manuscript, RFP and GFP are used respectively to label cancer and stromal cells. The resultant dual fluorescence in cancer-stromal fusion hybrids is evidence of the stromal contribution to cancer progression and metastasis.
- how about the tumor growth rate and ability of metastasis of these derivated clones of cancer-stromal cell? Also, did these colony AR dependent or AR independent?
Our response: We thanks the Reviewer’s insightful comments. Using the androgen receptor (AR) expressing LNCaP cells, we have previously shown that cancer-bystander cell fusion could produce AR-negative tumor cells (lines 77-79). In the “Limitations of the study” of the “Discussion” section, we pointed out the limitation of the current report (lines 483-490). As varied growth rates were observed from the 192 hybrid-derived clones during the continued culture (lines 280-309), we will present the growth rates in future reports in a quantitative fashion, together with individual clone’s AR status and metastatic potential.
- Tumor microenvironment is not only epithelial and stromal cells, there other cells like immune cells, macrophage, vessels cell should also discuss is the limitation part
Our response: We thank the Reviewer for this comment. In the revised manuscript, constituents of the tumor microenvironment have been outlined in the “Introduction” section (lines 65-67). In previous reports, we have studied cancer cell fusion with neural cells or bone marrow-derived mesenchymal stem cells of the tumor microenvironment. Main interest of the current study is in the cancer cell interaction with prostate-derived mesenchymal stromal cells.
